# Gated Calcium Ion Channel and Mutation Mechanisms in Multidrug-Resistant Tuberculosis

**DOI:** 10.3390/ijms24119670

**Published:** 2023-06-02

**Authors:** John A. D’Elia, Larry A. Weinrauch

**Affiliations:** Kidney/Hypertension Section, E P Joslin Research Laboratory, Joslin Diabetes Center, Department of Medicine, Beth Israel Deaconess Medical Center, Harvard Medical School, Boston, MA 02215, USA

**Keywords:** antibiotic resistance, calcium ion channels, extrusion of antibiotics, innate immunity, adaptive immunity, *Mycobacterium tuberculosis*

## Abstract

A wide spectrum of Gram-positive/Gram-negative bacteria has been found resistant to a wide spectrum of antibiotics in the United States of America during the past decade. Drug-resistant tuberculosis is not yet a major threat in North/South America, Europe, and the Middle East. However, the migration of populations in times of drought, famine, and hostilities may increase the global reach of this ancient pathogen. Given an increased spread from China and India to African countries, drug-resistant *Mycobacterium tuberculosis* has become an emerging topic of concern for Europe and North America. Due to the dangers associated with the spread of pathogens among different populations, the World Health Organization continues to expand healthcare advisories for therapeutic approaches for both stationary and migrating populations. As much of the literature focuses on endemic to pandemic viruses, we remain concerned that other treatable communicable diseases may be ignored. One such disease is multidrug-resistant tuberculosis. We focus on molecular mechanisms that this pathogen relies upon for the development of multidrug resistance via gene mutation and the evolutionary development of new enzyme and calcium channels.

## 1. Introduction

In a study based far from the borders or shores of the continental United States, a large spectrum of Gram-positive and Gram-negative bacteria has been found resistant to a large spectrum of antibiotics (as single or in combination). The incidence of infection with *Mycobacterium tuberculosis* was too low to document antibiotic sensitivities on a large scale [1]. At some point, migration of refugee families may provide sufficient clinical experience to add resistance to certain antibiotics (as single or in combination) by *Mycobacterium tuberculosis* isolates in the continental United States. During the first half of the first decade of 2000, India reported a high rate of infection with *Mycobacterium tuberculosis*. Several states, such as Gujarat, Maharashtra, and Andhra Pradesh, performed large-scale surveys which revealed multidrug resistance as high as 3% for newly diagnosed patients and 17% for patients previously treated for tuberculosis [2]. In the latter part of the first decade of 2000, these figures had already risen to 4% and 27%, respectively [3]. Findings in line with those in India were subsequently confirmed from high-density population centers in Africa [4]. In the first half of the second decade of 2000 in China, antibiotic resistance to *Mycobacterium tuberculosis* was as high as 3% for newly diagnosed patients and 20% for patient previously treated for tuberculosis [5,6]. The Beijing strain of *Mycobacterium tuberculosis*, most prevalent in China, is now recognized in India, Pakistan, Myanmar, and New York City. Due to its high level of infectivity and virulence, it is as prevalent as non-Beijing strains in Shanghai and Hong Kong. Non-Beijing strains have also been described in Taiwan, East Africa, East Africa/India, and Latin America/Mediterranean areas [7]. Up to 30% of transmission may occur from individuals with culture negative sputum, and what often appears to be a single geographic cluster of tubercular cases may represent several highly infectious individuals [8].

Concomitant risks to total body strength in some communities might be the result of simultaneous infestation with malaria or schistosomiasis. At this point in time, epidemiological studies of pulmonary infection are complicated by the tendency to explain outcomes according to positive tests for COVID virus, perhaps overlooking the presence of *Mycobacterium tuberculosis*. New instances of tuberculosis in Indonesia numbered 25/million/day, or approximately 7000 new tuberculosis cases/day within a population of 280 million people. At the peak of the COVID-19 pandemic, Indonesia was reporting new COVID-19 case rates of approximately 5000/day. Such numbers threatened to overwhelm the diagnostic and treatment capabilities of the healthcare system [9]. The host with debilitating infection from a parasite, virus, or bacterium complicates outcome studies of either one alone.
**1.** **Defining multidrug-resistant tuberculosis (MDR TB)**

MDR TB is caused by an organism that is resistant to at least isoniazid and rifampin, the two most potent TB drugs. Extensively drug-resistant TB (XDR TB) is a rare type of multidrug-resistant tuberculosis (MDR TB) that is resistant to isoniazid and rifampin, plus any fluoroquinolone and at least one of three injectable second-line drugs (i.e., amikacin, kanamycin, or capreomycin) [10].
**2.** **How and why does drug-resistant mycobacterium tuberculosis occur?**

Resistance occurs when prevalent *Mycobacterium tuberculosis* is exposed to and survive the encounter with therapy given inadequately (too short a duration, drug levels too low to kill, use of a single drug for treatment) or the mycobacterium is naturally cultured in populations with immune insufficiency resulting from HIV, coronavirus (MERS, SARS, COVID-19), or nutritional inadequacy. The molecular explanation for mutations that preserve the pathogenicity of *Mycobacterium tuberculosis* varies depending upon the substrate and environment in which it is cultured. Stopping MDR TB therapy during a viral plague, agricultural blight/drought, or in the presence of inadequate healthcare availability, frank poverty with changes in antibiotic regimens, or with the current slow development of new antitubercular agents may not be achievable. Drug tolerance may also occur in the absence of drug resistance-conferring mutations [11].
**3.** **Currently accepted antitubercular drug classes:**

Table 1a lists currently accepted antitubercular drugs. The mechanisms of antibiotic action against *Mycobacterium tuberculosis* are further identified in Table 1b.
**4.** **Are there clinical observations that could inform us of ways to increase effectiveness of antitubercular drug activity?**A decreased relative risk for new-onset *Mycobacterium tuberculosis* infection has been described among persons with hypertension treated with vs. without calcium channel blockers [12].A decreased mortality rate for subjects with both hypertension and tuberculosis treated with calcium channel blockers vs. those who did not receive calcium channel blockers [13]. Since some mechanisms of antimicrobial resistant mycobacterium involve construction of alternative non-calcium channels for extrusion of antibiotics, it is possible that adequate amounts of Vitamin D in macrophages may help keep calcium channels functioning in the bacterium. Calcium channels in the bacterium are inhibited by calcium channel blockers, resulting in retention of antibiotics. Retention of antibiotics allows for lower doses at lower cost.A decreased infection rate has been noted among individuals with diabetes mellitus immunosuppressed for kidney transplantation who did vs. did not receive calcium channel blockers for the treatment of hypertension [14]. Human macrophages laden with *Mycobacterium tuberculosis* have demonstrated a spontaneous capacity of the organism to avoid the bactericidal activity of isoniazid or rifampicin through the extrusion of antibiotics while still within the macrophage [15]. Following the addition of verapamil, the extrusion channel of *Mycobacterium tuberculosis* shuts down through loss of capacity to generate ATP. In addition, the macrophage was found capable of increasing both hydrolase enzyme activity and hydrogen ion concentration.A retrospective nested case-control analysis from a Taiwanese database identified a decreased risk of active tuberculosis among a population exposed to calcium channel-blocking agents when compared to a similar population not exposed to such medications [12]. 


**5.** 
**Can we interfere with the pathway by which *Mycobacterium tuberculosis* resists bactericidal activity, given that the mutations encountered may be different with respect to various antitubercular therapies?**



A wide diversity of mutations has permitted *Mycobacterium tuberculosis* to avoid specific fundamental host protections over decades of research. Increased pathogenicity occurs when bacterial mutation decreases uptake or increases the extrusion of an antibiotic. We must target methods to increase bacterial antibiotic uptake, perhaps through alternative delivery systems to affected organs, and to decrease bacterial extrusion of the antibiotic by interfering with ionic channel exchange. With respect to specific antitubercular therapies, we must consider some of the mechanisms of bacterial resistance:Aminoglycosides may be modified by *Mycobacterium tuberculosis* gene mutation for the reaction of hydroxyl and/or amino side chains, decreasing the efficacy of the antibacterial medication. The active aminoglycoside modifying enzymes include aminoglycoside acetyl transferase, aminoglycoside phosphotransferase, and aminoglycoside adenyl transferase [16].Isoniazid-resistant mutations of *Mycobacterium tuberculosis* include those that have lost peroxidase activity or the ability to form adducts with reduced nicotine adenine dinucleotide (NADH); or as sequential combinations of these three [17,18]. These mechanisms inhibit the construction of the wall of *Mycobacterium tuberculosis* [19]. Gene mutations from rifampicin-resistant *Mycobacterium tuberculosis* [12] have been identified [20]. Ethambutol-resistant *Mycobacterium tuberculosis* [4] isolated by DNA sequencing have demonstrated mutations in a specific area of the embCAB operon. These mechanisms inhibit the construction of the wall of *Mycobacterium tuberculosis* [17].Pyrazinamide-resistant *Mycobacterium tuberculosis* isolates have been commonly found. Under acid pH conditions, pyrazinoic acid inhibits the growth of mycobacterium tuberculosis [6].Specific regions of mutation (on the gyrA gene) of *Mycobacterium tuberculosis* have been noted to render the bacteria resistant to Levofloxacin over the past 20 years [21].Chemical structures of ethionamide and isoniazid are similar, resulting in mycobacterium resistance due to mutation-related loss in the activity of catalase and peroxidase, as well as the ability to form adducts with NADH. These mutations result in a thicker outer membrane of the pathogen with an increased capacity to create efflux pumps for the extrusion of ethionamide from resistant *Mycobacterium tuberculosis* [22].Injury to *Mycobacterium tuberculosis* from oxygen radicals generated by cycloserine may be altered by mutations in 16 different genes. The cell membrane focus involves the protection of phospholipids at the outer membrane site and quinones at the respiratory chain site of *Mycobacterium tuberculosis* [23,24,25].Para-aminosalicylic acid resistance of *Mycobacterium tuberculosis* involves genetic mutations in enzymes needed for the synthesis of folic acid [26,27].Linezolid-resistant *Mycobacterium tuberculosis* mutations occur in two separate specific genes (rplC and rrl genes) [28].*Mycobacterium tuberculosis* resistance to clarithromycin occurs through a gene encoding (ksgA) for RNA ribosomal dimethyl transferase enzyme [29].In order to minimize drug resistance to single antibiotic regimens in one gene site, a combination of isoniazid, ethambutol, and rifampicin has gained favor and shown successful results [30].


**6.** 
**Calcium channel extrusion of medications: mechanism of mycobacterium tuberculosis survival against higher doses of antibiotics.**



The thick outer envelope of *Mycobacterium tuberculosis* has two levels. The outer level is a lipid bilayer. The degree of attraction for the extrusion channel based on lipid solubility is associated with the degree of bactericidal activity. Antibiotics which have an affinity for lipid structures are most likely to open pores for infusion of other antibiotics that may not be as lipid avid. The outer membrane of *Mycobacterium tuberculosis* contains a channel protein (CpnT). This is the main protein for the structure involved in the extrusion of antibiotics. This CpnT protein is capable of mutation. Most studies look at the bacterium inside the macrophage. The outer membrane channel protein CpnT is useful for the intake of nutrients to *Mycobacterium tuberculosis*. Mutant groups bearing CpnT have been found to be 100 times more resistant to the uptake of antibiotics. Some studies identify this channel as calcium-independent [31,32]. Endotoxin from *Mycobacterium tuberculosis* can also stimulate a bronchoalveolar epithelial response stimulating the construction of an additional calcium channel (transient receptor potential (TRP) ion). This would be one of several levels of defense by which the epithelium activates a process of eliminating the bacterium. More importantly, the endotoxin activates macrophages capable of engulfing/digesting necrotic tissue and bacterium [33].

Laboratory studies have demonstrated the role of ion channels by which *Mycobacterium tuberculosis* extrudes bactericidal antibiotics and thus becomes resistant [34,35,36,37,38,39]. Although the tough outer membrane of *Mycobacterium tuberculosis* operates as a first line of defense, this structure is insufficient to protect the invading bacterium from a wide assortment of antibiotics gaining rapid entry. As a result, most research is directed at the capacity of *Mycobacterium tuberculosis* to extrude powerful antibiotics through channels in that tough outer membrane [33]. A single antibiotic may inhibit a single enzyme system in *Mycobacterium tuberculosis*. Given sufficient time, a single mutant strain of mycobacterium tuberculosis may avoid this enzyme inhibition. When a combination of two (or three) antibiotics is prescribed, the generation of two (or three) mutant strains that avoid two (or three) enzyme system inhibitions may take two (or three) times as much time. Although mutations can theoretically occur after *Mycobacterium tuberculosis* is engulfed by macrophages, the time interval before injury by digestion is short. A mechanism of rapid extrusion of antibiotics is needed. Protein-derived calcium ion channels are a rapid mechanism by which *Mycobacterium tuberculosis* extrudes antibiotics, whether free in the bronchioalveolar epithelium or engulfed in macrophage cells. Calcium channel blockers may keep the concentration of antibiotics high enough to injure *Mycobacterium tuberculosis* before the pathogen has had sufficient time to mutate to a state of antibiotic resistance (Table 2). Whereas clearance of lithium increases and cyclosporine decreases in the presence of systemic calcium channel blockade, we have no current knowledge of any changes in the clearance of antituberculosis medications during the use of calcium channel blockers [35,36]. Ten years after observation of increased lithium clearance in bipolar patients commenced on verapamil for hypertrophic cardiomyopathy, the mechanism of increased clearance was elaborated in detail [37]. Calcium channel blockers were found to be antihypertensive not just due to large vessel vasodilation but also due to dilation of the afferent arteriole of the renal glomerulus resulting in increased tubular filtration. An additional inhibitory response to the mineralocorticoid receptor in the distal tubule similar to the effects of spironolactone or eplerenone was described. This natriuretic effect also included increased lithium clearance [37].

Other considerations regarding calcium blockade for patients infected with *Mycobacterium tuberculosis* include: 1. Retention of beta cell secretion in early-onset type 1 diabetes via protection from intracellular calcium toxicity-related apoptosis [38,39]; 2. Reduction of calcineurin inhibitor requirement in solid organ allograft recipients; and 3. Enhanced clearance of single charged cations in patients dependent upon lithium therapy (Table 3).

Rifampicin-resistant *Mycobacterium tuberculosis* has been shown to extrude rifampicin through channels which have some characteristics of calcium channels [40]. The extrusion may be blocked by verapamil, after which there is improved efficiency of the extruded antibiotic [41]. Ion channels involved in antibiotic resistance through blockade of the extrusion channel may [42] or may not be a standard calcium channel [43]. Some human studies indicate that dihydropyridine calcium channel blockers may be at least as effective as verapamil, the more commonly studied calcium channel blocker in experimental studies [44]. The most astounding discoveries have been that of a new class of channel-forming outer membrane proteins [45] where de-novo toxins cause sufficient necrosis to enlarge the pathway for extrusion [46].
**7.** **Intrinsic resistance to mycobacterium tuberculosis by the human organism**

Resistance to mycobacterial invasion increases in the presence of an intact homeostatic human organism. Inadequate rest, nutritional or hygienic resources, or relative immunosuppressive states brought about by comorbid states or therapies increase the likelihood of failure to adequately contain or localize mycobacterial infection. In some populations, defenses against mycobacterium tuberculosis are impaired by genetic traits involving ion channels. Populations receiving immunosuppressive therapy, such as tumor necrosis factor-alpha (TNF) antagonists, systemic corticosteroids, or immunosuppressive drug therapy following organ transplantation, are at the greatest risk for active tuberculosis.

Study subjects with diabetes mellitus (with or without complications) have a significantly lower risk of *Mycobacterium tuberculosis* infection when treated with calcium channel blockers (dihydropyridine or non-dihydropyridine) as opposed to study subjects with diabetes mellitus not treated with calcium channel blockers [42]. A specific calcium channel receptor (P2X7) is upregulated in persons with diabetes mellitus and tuberculosis [47].

In obesity/type 2 diabetes mellitus, the increase in several components of natural and adaptive immunity is associated with insulin resistance. At the same time, a reduction in the number of regulatory T lymphocytes and natural killer cells, as well as the dysfunction of macrophages, renders the host more susceptible to infection [48].

In response to pathogens, immune cells may change concentration of calcium ions by opening channels from outside the cell [49] or from opening of channels from the endoplasmic reticulum inside the cell [41]. Cells from immune/inflammasome cascades participate [46], including innate (dendritic cells, macrophages) and adaptive cells (T lymphocytes, B lymphocytes). 

With respect to innate immunity, dendritic cells ingest *Mycobacterium tuberculosis* and forward the antigen signal to uncommitted lymphocytes [50,51,52]. Mycobacterium may be strong enough, however, to injure dendritic cells. Macrophages ingest *Mycobacterium tuberculosis* more aggressively than dendritic cells, but they are less efficient in forwarding antigen signals to uncommitted lymphocytes. Mycobacterium may be strong enough to injure macrophages as well [52,53,54,55]. Differentiation of monocytoid cells into macrophages and migration of macrophages after stimulation have an association with L-type calcium channels. Both amlodipine and verapamil were shown to interfere with differentiation and migration of macrophages [55]. Human dendritic cells express L-type calcium channels supporting two dendritic cell functions. The first involves engulfing pathogens. The second involves expression of Interleukin 12. When an invasive virus attempted to shut down this L-type calcium channel, a dihydropyridine agonist was found to retain function. The virus and dihydropyridine calcium channel protector drugs were shown to compete for access to dendritic cells [56].

With respect to adaptive immunity, uncommitted lymphocytes are transformed into T lymphocytes. CD4+ T lymphocytes generate cytokines designed for the elaboration of inflammation cascades against mycobacterium tuberculosis. Uncommitted lymphocytes are transformed into B lymphocytes which generate antibodies against *Mycobacterium tuberculosis* as a major function with a minor function of generating cytokines for inflammation cascades. (Table 4, Figure 1).
**8.** **Co-infection with mycobacterium tuberculosis and COVID-19 in high-risk groups: calcium channel mechanisms leading to toxic accumulation of calcium**

Clinical observations of treatment with calcium channel blockers to control hypertension gain potential insight into calcium channel mechanisms in infection with or without *Mycobacterium tuberculosis* in study cohorts with or without diabetes mellitus. It is important to note that both COVID-19 virus and multidrug-resistant *Mycobacterium tuberculosis* are spread by droplets. Infectious virulent strains spread with population mobility from areas with highest endemic prevalence in Eastern Europe and central Asia to areas with lower prevalence [57]. Confusion over chest radiographic findings of COVID and mycobacterium in areas where laboratory identification of bacteria and/or virus may not be easily available is a problem. Ten million new instances of tuberculosis were reported in 2018, the year before the onset of the COVID pandemic [58]. Co-infection of eighty individuals with *Mycobacterium tuberculosis*/COVID-19 was first documented in 2020 in Southern Europe in migrating males [59]. Subsequently, a 2022 global survey found a tenfold increase of patients with co-infection. The mortality rate in migrating males was 14% vs. 9% for the other patients who were not migrating [60]. 

Diabetes mellitus accounted for 24% of the major underlying risk factors in this co-infection study group. Defects in insulin secretion render certain persons with type 2 diabetes mellitus more susceptible to the complications of uncontrolled hyperglycemia. A study from Northern Europe has identified a mutation in the L-type gated calcium channel closely connected with Beta cell insulin secretion that would be expected to diminish the response to elevated blood glucose [61] and potentially render these individuals more susceptible to viral infection with or without *Mycobacterium tuberculosis* co-infection.

Results comparing normal controls to an animal model of type 1 diabetes mellitus (streptozotocin diabetic mouse) were useful in clarifying the risk of pulmonary cavitation after an aerosol of mycobacterium tuberculosis. The cohort with diabetes mellitus demonstrated a higher burden of macrophages laden with *Mycobacterium tuberculosis* bacilli as well as delayed accumulation of white blood cells at the site of infection [62]. 

In Shandong, China, 426 patients with diabetes mellitus demonstrated a higher incidence of pulmonary cavitation at the time of primary diagnosis of tuberculosis compared to 6797 patients who were not diagnosed with diabetes mellitus at the time of primary diagnosis of tuberculosis [63]. For persons with diabetes mellitus infected with *Mycobacterium tuberculosis*, an increase in the expression of interleukins with activation of macrophages is the defense process initiated by the onset of pulmonary cavitation [64]. However, when pulmonary alveolar macrophages from individuals with diabetes mellitus were infected with *Mycobacterium tuberculosis* bacilli, defense mechanisms such as the expression of interleukins were slower in process compared to alveolar macrophages from individuals not diagnosed with diabetes mellitus [65]. For persons with diabetes mellitus infected with COVID-19, acute respiratory distress syndrome is associated with the release of intracellular calcium stores in the endoplasmic reticulum, allowing toxic accumulation in the cytoplasm [66]. For persons with co-infection *Mycobacterium tuberculosis*/COVID19, acute lung injury may also result in calcium injury following entry through a non-L-type gated calcium channel. This store-operated calcium channel controls entry of calcium from outside the cell. An experimental rat model of lung injury following hemorrhagic shock used the polymorpho-nuclear white blood cell (neutrophil) to demonstrate calcium inflow from outside the cell. In this experimental model, the toxic accumulation of calcium was prevented by the use of a dihydropyridine calcium channel blocker [66]. 

It has been known that rifampicin kills rapidly growing Mycobacterium tuberculosis pathogens. A World Health Organization report stated there are nearly 500,000 new instances of multidrug-resistant *Mycobacterium tuberculosis* infections per year [67]. In 2019, a 6-month study found the combination of the new drug pretomanid + bedaquiline + linezolid to be useful not only for rapidly growing isolates but also for those which persist with less potent drug combinations [63].

A specific concern with the use of a three-drug combination taken by mouth would be the effect on the gut microbiome. A study of *Mycobacterium tuberculosis* susceptible to the three-drug combination identified a gene (PE-PGR557) not present in members of the microbiome. Thus, there was no change in the bacterial content of gut bacteria [68].

Vitamin D, vitamin D receptor, and vitamin D binding protein deficiencies exert limitations on resistance to *Mycobacterium tuberculosis* infection, as discussed in a noteworthy review [69]. Vitamin D binding protein is particularly relevant as it increases availability through reabsorption in the proximal tubule of vitamin D filtered by the kidney glomerulus. The nuclear receptor for 25 (OH) vitamin D enhances the expression of peptides (cathelicidin, defensin) with anti-microbial function. Inhibition of matrix metalloproteinase, which degrades bronchopulmonary tissue ahead of invasion by *Mycobacterium tuberculosis*, is a direct benefit of 25 (OH) vitamin D. An experimental model has demonstrated that 1,25. dihydroxy vitamin D activates L-type voltage-gated calcium channels through a mechanism not involving nuclear gene regulation [70]. Since some mechanisms of antimicrobial-resistant *Mycobacterium tuberculosis* involve the construction of alternative channels for the extrusion of bactericidal antibiotics, it is possible that adequate amounts of vitamin D for *Mycobacterium tuberculosis* would be useful in maintaining standard calcium channels that can be inhibited by calcium channel blockers [71].
**9.** **To what extent does in vitro laboratory testing remain useful in directing therapy, or must we develop new means of testing in cases of MDR tuberculosis?**

Culture remains the gold standard for TB diagnosis and drug-resistance testing. Due to the slow growth of *Mycobacterium tuberculosis,* testing takes a considerable time. There are tests (individual drug susceptibility) to detect resistance to certain antibiotics based on molecular genetic evaluations. There are currently no tests that can be used to assess host immune competence. There are also no methods to test the effect of complementary measures to buttress host defense against mycobacterium or augmentation of the bactericidal effect of antibiotics. To date, there is no data on whether slowing of growth in culture by inhibition of ion channels can be used to predict bactericidal effectiveness.

## 2. Conclusions

The major genetic, biochemical, and biophysical mechanisms that have enabled mycobacterium tuberculosis to survive through the past century of antibiotics have been reviewed. Through many centuries, individuals immunocompromised by malnutrition have not been able to eradicate or accommodate this pathogen by natural (innate + adaptive) immunity due to immunosuppression associated with fatigue (long hours of work), poor hygiene (water shortage), and malnutrition (crop failure). In the past few years, when perhaps one-fourth of the world’s population struggles with this bacterial risk to survival, the situation has been exacerbated by populations displaced by hostilities, natural catastrophes, and a superimposed COVID-19 viral pandemic. It remains to be seen whether international cooperation can ameliorate shortages of healthcare, medication, food, shelter, and fuel to overcome the spread of multidrug-resistant tuberculosis. In the past 50 years, the world’s population has struggled with viral diseases (hepatitis, HIV, Ebola, Marburg, neuro, monkeypox, COVID) plus the simultaneous spread of antibiotic-resistant tuberculosis, which together are capable of overwhelming public health facilities in dense populations of Asia and Africa. The migration of men looking for work will eventually bring this challenge to continental Europe, the United Kingdom, and North and South America. International cooperation to ameliorate shortages in medication, food, and shelter will be needed.

## Figures and Tables

**Figure 1 ijms-24-09670-f001:**
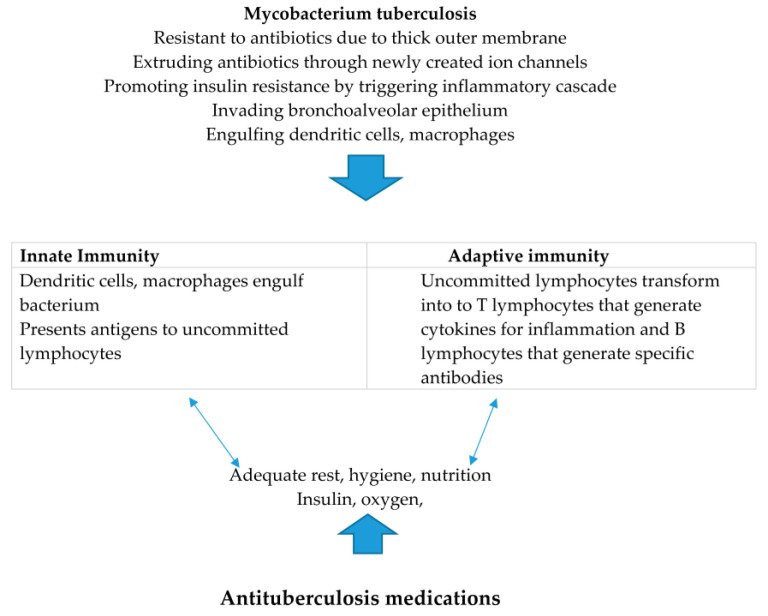
Drug-resistant tuberculosis infections.

**Table 1 ijms-24-09670-t001:** (**a**) Currently accepted antitubercular drug classes. (**b**) Antibiotic action against *Mycobacterium tuberculosis*: resistance through identified mutations.

(**a**)	
Group 1	isoniazid, rifampicin, ethambutol, pyrazinamide, rifabutin
Group 2	(parenteral) kanamycin, amikacin, capreomycin, streptomycin
Group 3	fluoroquinolones (moxifloxacin, levofloxacin, ofloxacin)
Group 4	(bacteriostatic) ethionamide, protionamide, cycloserine, terizidone, para-amino salicylic acid
Group 5	(data pending) clofazimine, linezolid, amoxicillin/clavulanate, thioacetazone, imipenem/cilastatin, high-dose isoniazid, clarithromycin
(**b**)	
**A.** **Mutation sites with biochemical identification** **a.** Acid pH inhibition of growth **Pyrazinamide** **b.** Oxygen radical peroxidation of phospholipids **Cycloserine** **c.** NADH/Quinones: interfering with electron transport chain/high energy phosphate transfer (ATP) **Ethionamide, Isoniazid** **d.** RNA ribosomal transferases Dimethyl transferase (ksgA gene encoding) for a ribosomal RNA) promotes growth proteins **Clarithromycin** Acetyl transferase, Phosphotransferase, Adenyl transferase inhibition of generation of growth protein **Streptomycin** **e.** Folic acid synthesis perturbation by mutations in thymidylate synthase A and dihydrofolate reductase **Para-amino salicylic acid** **B.** **Mutation Sites without biochemical identification** **Rifampicin**—95% within resistance-determining region **Ethambutol**—79% within resistance-determining region **Levofloxacin**: gyrA gene **Linezolid**: rplC and rrl genes **C.** **Unknown Site or Reactions** **Amoxicillin/Clavulanate**: none yet identified **Carbapenems**: none yet identified **Imipenem/Cilastatin**: none yet identified	

**Table 2 ijms-24-09670-t002:** Calcium-related mechanisms of Survival of *Mycobacterium tuberculosis*.

1	Injury to neutrophils by accumulation/redistribution
	a. From external sources via calcium channels *
	b. From internal storage areas via calcium channels
2	Extrusion of antibiotics from *Mycobacterium tuberculosis*
	a. From internal locale via calcium channels *

* May be inhibited by calcium channel blockers.

**Table 3 ijms-24-09670-t003:** Multiple effects of calcium channel blocking medications.

**1**	Vasodilation of afferent arteriole of kidney glomerulus
**2**	Loss of sodium, lithium from kidney distal tubule
**3**	Blockade of channels by which bacterial pathogens extrude antibiotics
**4**	Blockade of entry of calcium into cells
	a. for a calcineurin cascade to increase nuclear factor kappa beta inflammation
	b. for Beta cell toxicity in new-onset Type 1 diabetes

**Table 4 ijms-24-09670-t004:** Balance of attack/defense mechanisms of *Mycobacterium tuberculosis* and the host.

Pathogen Attack	Host Defense
Dislodging of bronchoalveolar epithelium	Nutrition, oxygen, rest
Engulfment of dendritic cells, macrophages	Innate immunity: engulfment of MTB by dendritic cells, macrophages followed by activation of T and B lymphocytes
	Adaptive immunity: T lymphocytes release cytokines (a non-specific inflammatory reaction) and B lymphocytes create antibodies (for MTB specific response)
Thick outer membrane blocks drug entry	Antibiotic
Develops ion channels for extrusion of antibiotics	CCB inhibits extrusion of antibiotics eliminating current strain of MTB
Mutated populations evolve drug resistance as life span increases	Potential target for research

MTB = Mycobacterium tuberculosis. CCB = calcium channel blockers.

## Data Availability

Not applicable.

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
