# Peer review of "Gated Calcium Ion Channel and Mutation Mechanisms in Multidrug-Resistant Tuberculosis"

_ijms, 2023, doi:10.3390/ijms24119670_

Round 1
Reviewer 1 Report
Dear Authors,
thank you for the opportunity to read your review. Unfortunately, the review contains many major errors. It is also not written in accordance with the MDPI requirements (even citations that are placed in parentheses).
Errors that must be corrected:
- Gram-positive, Gram-negative - "Gram" is written with a capital letter, this name comes from the surname. Please correct for the correct entry in the entire review
- Citations should not be used in the Abstract. It is to be written from you and contain a summary of the entire review. Please change it, rewrite it
- line 9 - why is "drug" capitalized?
- the name of the bacteria, in this case Mycobacterium tuberculosis, is written in italics (e.g. line 26-27), and the first part is capitalized (e.g. line 51). Change in the whole review.
- line 52 - "mycobacteria" is laboratory jargon, not used in publications. Please change the review to more scientific, publishing language
- too few items cited. There are only 65 of them, which definitely reduces the quality of the presented review. The review should be an exhaustive discussion. Unless it covers a specific period, which the Authors have not indicated in the text
The subject of the review is interesting and "up-to-date", but the form in which it is presented is unassimilable by the reader, and therefore impossible to accept. Please change this form.
The overview would be more attractive if it included tables, diagrams. Couldn't chapter 7 be presented in the form of a diagram? include in it what dose, what drug and what effect on the survival of M. tuberculosis? And Chapter 6, perhaps, could be presented in the form of a table? This is only a proposal to consider, but maybe the authors will see other possibilities.
A very important piece of information that was missing from this review is how resistant strains are transmitted. What is the epidemiology? Which countries have the highest number of cases and how these strains spread during the mobility of citizens. Unfortunately, the authors omitted this, and such information would be very valuable and would definitely increase the value of this review. I would be happy to reconsider this review after making the above changes.
Best regards
Author Response
See attached responses for Reviewer 1.

Reviewer 2 Report
The authors presented a Review Paper entitled "GATED ION CHANNEL AND MUTATION MECHANISMS IN MULTIDRUG RESISTANT TUBERCULOSIS". In general is a well-written document but it seems to lack scientific rigor.
There are some observations below;
Major:
- The theme is very important so I suggest revising the information on the field and generating a Systematic Review and Meta-Analysis.
-L45. The definition of MDR-TB is not supported by an adequate literature review.
- When you mention resistance mechanisms, you refer in an important way to mutations (undoubtedly the most frequently reported) but you are leave out the acquisition of resistance mechanisms. Are there no reports?
- You are not considering Mtb tolerance.
-Tables 1, 2 and 4 are poorly formatted, lacking of an adequate caption.
-Fig 1 is not well formatted.
Minor:
- The information between lines 62-69 would fit better in a table. It gives the impression of having been carelessly placed.
- L72, Before starting the information in the paragraph it would be convenient to mention something that introduces us to it. .
-L253, I am unable to identify the association of MDR-TB with COVID, without sufficient data COVID could be associated with any other infection.
Author Response
See attached responses for Reviewer 2.

Round 2
Reviewer 1 Report
Dear Authors,
Thank you for the corrections made. Most of them are acceptable to me. As for the use of naming, I know that it should be in italics, but I will leave the decision to the Editors.
Best regards
Author Response
Thank you for your comments. I have changed to italics for Mycobacterium tuberculosis.
Reviewer 2 Report
The authors have subbmitted a revised version of the manuscript entitled "GATED ION CHANNEL AND MUTATION MECHANISMS IN MULTIDRUG RESISTANT TUBERCULOSIS" several observations were addressed but there are some minor revisions still needed.
- Tables are no formatted, they seems to be lists. The MDPI template harbors a table format, I suggest to use it for fitting the information.
- According with microbial nomenclature, all names must be italicized. If it is the first time you mentioned a microorganism name, you should write it complete, after you can abbreviate. All microorganism names initial word must be written in capital letter, the remaining words lowercase. Always italicized, including in titles. If you are mentioning only genera (e.g. Mycobacterium) should be written italicized and with the initial capital letter. When you mentioned "tuberculosis" if you are referring to the illness it is not necessary to be italicized, but if you are referring to the bacterial specie then it must be italicized. These conventions are independent on the literature consulted.
Author Response
Thank you for your comments, I have made the appropriate changes.
